# Investigation into the Optimal Strategy of Radium-223 Therapy for Metastatic Castration-Resistant Prostate Cancer

Yasuo Oguma [1,*], Makoto Hosono [2,*], Kaoru Okajima [1], Eri Inoue [1], Kiyoshi Nakamatsu [2], Hiroshi Doi [2], Tomohiro Matsuura [2], Masahiro Inada [2], Takuya Uehara [2], Yutaro Wada [2], Aritoshi Ri [2], Yutaka Yamamoto [3], Kazuhiro Yoshimura [4], Hirotsugu Uemura [4] and Yasumasa Nishimura [2]

[1]  Department of Radiology, Nara Hospital, Kindai University, Nara 630-0293, Japan
[2]  Department of Radiation Oncology, Kindai University Faculty of Medicine, Osaka 589-8511, Japan
[3]  Department of Urology, Nara Hospital, Kindai University, Nara 630-0293, Japan
[4]  Department of Urology, Kindai University Faculty of Medicine, Osaka 589-8511, Japan
*  Correspondence: oguma@med.kindai.ac.jp (Y.O.); hosono@med.kindai.ac.jp (M.H.)

**Simple Summary:** The optimal combination and sequence of treatments for castration-resistant prostate cancer were investigated by identifying prognostic factors after Ra-223 administration and inferring their causal relationship. We found that various biomarkers and factors are involved in prognosis, among which early administration of Ra-223 (up to the third line) was associated with better prognosis and prolonged time to pain worsening.

**Abstract:** The optimal sequence and combination of radium-223 therapy (Ra-223) for castration-resistant prostate cancer with bone metastasis (mCRPC) remain unclear. This study aimed to explore the prognostic factors after Ra-223 administration and to determine the optimal treatment strategy. We enrolled 64 patients with mCRPC who underwent Ra-223 therapy from June 2016 to July 2022 at a single institution in Japan. Overall survival (OS) and pain progression-free survival (p-PFS), which was proposed as a measure of quality of life (QOL), were analyzed using Cox proportional hazards models and log-rank tests, and between-factor analysis was performed with the Mann–Whitney U (MWU) test. Univariable and multivariable analyses revealed prognostic factors; specifically, early treatment (≤third line), completion of six treatment cycles, low bone scan index (BSI) (<0.61), alkaline phosphatase (ALP) (<140 U/L), prostate-specific antigen (PSA; <22.9 ng/mL), lactate dehydrogenase (LDH; <240 U/L), high hemoglobin (Hb) (≥11.4 g/dL), and prior denosumab use significantly prolonged OS. Low BSI, low ALP, and early Ra-223 treatment also prolonged p-PFS in the log-rank tests. The MWU test showed that high BSI (≥0.61) was associated with high PSA and high ALP and a tendency for Hb to decrease. Late Ra-223 treatment (≥fourth line) was significantly associated with low Hb and high PSA. Early Ra-223 treatment was significantly associated with improved OS, and administering Ra-223 before novel hormonal or anticancer agents may be meaningful.

**Keywords:** castration resistant; prostate cancer; bone metastasis; radium-223; abiraterone; enzalutamide; bone scan index; bone scintigraphy; quality of life; palliation

## 1. Introduction

The prognosis of patients with metastatic prostate cancer is approximately 42 months [1], and the patient often eventually develops bone metastases, with autopsy results showing bone metastases in over 90% of cases [2]. Bone metastases are associated with symptomatic skeletal events (SSEs) and reduce the quality of life (QOL). Therefore, the management of bone metastases is important. Since the 2010s, drug therapy options for castration-resistant prostate cancer with bone metastasis (mCRPC) have increased from docetaxel [3], cabazitaxel [4], and novel hormonal agents, such as abiraterone [5] and enzalutamide [6], to alpha-particle-emitting radium-223 (Ra-223) therapy in 2013. Radium is comparable to

calcium in that both are alkaline earth metals. With Ra-223 therapy, radium is taken up by the osteogenic metastases of prostate cancer instead of calcium and emits alpha rays from within the bone metastasis, causing a double-strand break in the DNA. The in-vivo range of alpha particles is less than 100 μm, which in cellular size is expected to efficiently affect metastases while minimizing the effect on normal bone marrow [7,8]. The efficacy and safety of this drug in terms of overall survival (OS) and QOL were demonstrated in an international phase III study [9,10], which showed a significant prolongation of median OS (14.9 months vs. 11.3 months) and time to first SSEs (15.6 months vs. 9.8 months) compared with the placebo. Ra-223 was approved in the USA and Europe in 2013 and in Japan in 2016. Studies on this treatment have shown that various biomarkers [11–15], number of doses [16,17], prostate-specific antigen (PSA) doubling time [14,18,19], and timing of administration [20,21] are related to prognosis. Although the risks of individual biomarkers have gradually emerged, to the best of our knowledge, the causal relationship between each factor, optimal sequence, and combination of Ra-223 and other therapies remain unclear. This study aimed to explore the prognostic factors after Ra-223 administration and to determine the optimal treatment strategy.

## 2. Materials and Methods

### 2.1. Study Design

This study was approved as a single-institution, retrospective study by the Institutional Review Board of the Kindai University Faculty of Medicine, which included Kindai University Hospital and Nara Hospital, Kindai University.

### 2.2. Study Patients

From June 2016 to July 2022, 64 patients with mCRPC who received more than one dose of Ra-223 therapy at Kindai University were enrolled. All patients were pathologically diagnosed with prostate adenocarcinoma by prostate needle biopsy. Gleason scores were evaluated using the grading system proposed by the 2014 International Society of Urological Pathology (ISUP) [22]. Castration resistance was defined based on an elevated serum PSA level despite continued androgen deprivation therapy and was ultimately determined clinically by a urologist. All bone metastases were detected using bone scintigraphy, and the bone scan index (BSI) and magnetic resonance imaging (MRI) were also used as adjunctive diagnostic tools when applicable. BSI is a new evaluation index of bone scintigraphy [15]. BSI values were calculated using a Neural Network-based diagnostic support software BONENAVI, version 2 (FUJIFILM RI Pharma Co.; Tokyo, Japan). None of the patients had organ metastasis at the first administration of Ra-223, and 32 patients had lymph node metastasis. All patients received Ra-223 at a dose of 55 kBq/kg intravenously every 4 weeks for a maximum of six doses.

### 2.3. Data Collection

All medical information was retrospectively collected from clinical records. Blood chemistry (lactate dehydrogenase (LDH), alkaline phosphatase (ALP), calcium, albumin), PSA levels, and blood counts were measured. Blood was drawn from all patients within 10 days before each Ra-223 administration (described as the "baseline" value before the first dose) and within 1 month after the last dose (last value). The difference rate in percentage (described as "_rate") for each biomarker was defined as (last value-baseline value)/baseline value. PSA doubling time (PSADT) was determined by exponentially approximating at least four PSA data points showing monotonic increases at 2 months before and after the first dose of Ra-223 using Microsoft Excel and dividing log(2) by the obtained coefficient (days) on time. Since the measurement method for ALP was changed to that used by the International Federation of Clinical Chemistry (IFCC) from that used by the Japanese Society of Clinical Chemistry (JSCC) in April 2020, all data collected by the JSCC were unified to the IFCC by a conversion formula, and the value of ALP obtained from the JSCC was divided by 2.84 [23]. Serum calcium (mg/dL) was corrected for albumin

levels by adding 4 to the measured value of calcium and subtracting the serum albumin value. Neutrophil-to-lymphocyte ratio (NLR) was calculated by dividing the neutrophils count by the lymphocytes count. Pain progression-free survival (p-PFS) was defined as the time from the day of the first dose injection of Ra-223 to the day of pain progression. Here, pain progression was defined as the initiation of analgesics or opioids.

### 2.4. Statistical Analysis

Statistical analysis was performed using R software version 4.0.5 (The R Foundation for Statistical Computing, Vienna, Austria) [24] and EZR software (Jichi Medical University, Saitama, Japan). The Kaplan–Meier method was used for OS and p-PFS. OS was evaluated from the day of the first dose injection or the first day of the entire treatment after diagnosis of mCRPC to the day of follow-up or death. If a patient's follow-up was censored by transfer for terminal care, the date of death was set to 2 months after that date. Each prognostic factor was estimated using the log-rank test and Cox proportional hazards model. Two-group comparisons between prognostic factors were performed using the Mann–Whitney U test. Correlations between two variables were analyzed using Spearman's rank correlation coefficient. Differences between the two groups were considered significant with a $p < 0.05$. The optimal cut-off value for each factor analysis was determined from the receiver operating characteristic (ROC) curve for death.

### 3. Results

#### Characteristics and Clinical Outcome of Ra-223 Therapy

All the characteristics of the patients are shown in Table 1. The median follow-up time from the first dose of Ra-223 was 10.9 (1.2–73.6) months. Median OS and p-PFS were 24.0 (13.4–32.6) and 27.9 (18.3-NA) months, respectively. Of the 64 patients, 24 (37.5%) had pain symptoms and were receiving analgesics or opioids at the time of the first administration of Ra-223. Univariable and multivariable analyses were performed using the log-rank test and Cox proportional hazards model to explore the prognostic factors (Table 2). Among the factors analyzed, the area under the ROC curve for death was the highest for baseline BSI at 0.704. The Youden index and the left upper angle nearest point of the ROC curve of the baseline BSI were both 0.61, which was defined as the cutoff point (Figure 1).

**Table 1.** Characteristics of patients.

| Characteristics | Median | Minimum | Maximum |
|---|---|---|---|
| Follow-up time, median (range) (month) | 10.9 | 1.2 | 73.6 |
| Age, median (range) | 74 | 49 | 87 |
| Height, median (range) (cm) | 164 | 147 | 177 |
| Weight, median (range) (kg) | 60 | 37 | 99 |
| T stage | | | |
| T1 | 1 | 2.0% | |
| T2 | 11 | 22.0% | |
| T3 | 28 | 56.0% | |
| T4 | 10 | 20.0% | |
| Not available | (14) | | |
| Gleason score (GS), n (%) | | | |
| GS6 | 1 | 1.6% | |
| GS7 | 9 | 14.8% | |
| GS8 | 8 | 13.1% | |
| GS9 | 31 | 50.8% | |
| GS10 | 12 | 19.7% | |
| Not available | (3) | | |

**Table 1.** *Cont.*

| Characteristics | Median | Minimum | Maximum |
|---|---|---|---|
| Lymph node metastasis, n (%) | | | |
| Yes | 32 | | 56.1% |
| No | 25 | | 43.9% |
| Not available | (7) | | |
| Number of doses, n (%) | | | |
| 1 | 1 | | 1.6% |
| 2 | 6 | | 9.4% |
| 3 | 7 | | 10.9% |
| 4 | 3 | | 4.7% |
| 5 | 5 | | 7.8% |
| 6 | 42 | | 65.6% |
| Treatment line of Ra-223, n (%) | | | |
| Early (≤3) | 37 | | 57.8% |
| Late (≥4) | 27 | | 42.2% |
| Pain * at the first dose of Ra-223 administration | | | |
| Yes | 24 | | 37.5% |
| No | 40 | | 62.5% |
| PSA * at diagnosis (iPSA) | 195.2 | 4.8 | 6453.6 |
| PSA at baseline | 22.9 | 0.0 | 2193.0 |
| PSA_rate | 116% | −100% | 8332% |
| Increase | 48 | | 60.0% |
| Decrease | 12 | | 20.0% |
| PSADT * (days) | 58.7 | 24.0 | 608.2 |
| LDH at baseline (U/L) | 213.5 | 149.0 | 1720.0 |
| LDH_rate | 3.7% | −31.2% | 914.4% |
| ALP at baseline (U/L) | 113.6 | 35.0 | 1296.8 |
| ALP_rate | −16.7% | −92.0% | 1322.3% |
| Increase | 18 | | 28.6% |
| Decrease | 45 | | 71.4% |
| ALP_rate | −16.9% | −92.0% | 91.1% |
| Calcium (albumin-corrected) (mg/dL) | 9.4 | 8.0 | 13.0 |
| Ca_rate | 0.5% | −33.1% | 19.0% |
| Alb (g/dL) | 3.8 | 2.8 | 4.6 |
| Hb (g/dL) | 12.0 | 8.8 | 15.4 |
| Hb_rate | −7.4% | −61.8% | 19.6% |
| Bone scan index (BSI) | 0.93 | 0.00 | 10.32 |
| Prior use of taxane, n (%) | | | |
| Yes | 32 | | 50.0% |
| No | 32 | | 50.0% |
| Prior (or concurrent) use of abiraterone, n (%) | | | |
| Prior | 42 | | 65.6% |
| Concurrent | 3 | | 4.7% |
| No | 19 | | 29.7% |
| Prior use of enzalutamide, n (%) | | | |
| Yes | 43 | | 67.2% |
| No | 21 | | 32.8% |
| Prior use of bone supportive agent, n (%) | | | |
| Bisphosphonate | 6 | | 10.0% |
| Denosumab | 36 | | 60.0% |
| No | 18 | | 30.0% |
| Not available | (4) | | |

Pain *: use of analgesics or opioid; baseline: the value of the first dose of Ra-223 administration; PSA: prostate-specific antigen; PSADT: PSA doubling time; ALP: alkaline phosphatase; LDH: lactate dehydrogenase; Ca: calcium; Alb: albumin; Hb: hemoglobin; Neut: neutrophil; Plt: platelet; "_rate": difference rate determined from the formula ((last value-baseline value)/baseline value)

**Table 2.** Result of Cox proportional hazards analysis for OS.

| | Univariable | | | | Multivariable | | | |
|---|---|---|---|---|---|---|---|---|
| | HR | 95% CI | | *p* | HR | 95% CI | | *p* |
| Age $\geq$ 76 (vs. <76) | 0.905 | 0.457 | 1.793 | 0.775 | | | | |
| Weight $\geq$ 63 (vs. <63) (kg) | 0.764 | 0.381 | 1.532 | 0.448 | | | | |
| BSI $\geq$ 0.61 (vs. <0.61) | 4.805 | 1.918 | 12.040 | <0.001 | 2.848 | 0.846 | 9.591 | 0.091 |
| Number of doses, 6 (vs. $\leq$5) | 0.181 | 0.088 | 0.372 | <0.001 | 0.155 | 0.026 | 0.943 | 0.043 |
| Treatment line of Ra-223, Late ($\geq$4) vs. Early ($\leq$3) | 2.432 | 1.246 | 4.746 | 0.009 | | | | |
| GS = 10 (vs. $\leq$9) | 5.043 | 1.855 | 13.700 | 0.002 | | | | |
| T4 (vs. $\leq$3) | 2.193 | 0.984 | 4.886 | 0.055 | | | | |
| iPSA $\geq$ 177 (vs. <177) (ng/mL) | 1.129 | 0.575 | 2.220 | 0.724 | | | | |
| PSA $\geq$ 22.9 (vs. <22.9) (ng/mL) | 4.785 | 2.208 | 10.370 | <0.001 | 2.435 | 0.887 | 6.681 | 0.084 |
| PSA_rate $\geq$ 45 (vs. <45) (%) | 5.858 | 2.504 | 13.700 | <0.001 | 6.052 | 1.671 | 21.910 | 0.006 |
| PSADT $\geq$ 90 (vs. <90) (days) | 0.397 | 0.193 | 0.820 | 0.012 | | | | |
| Alb $\geq$ 3.8 (vs. <3.8) (g/dL) | 0.369 | 0.185 | 0.735 | 0.005 | | | | |
| ALP $\geq$ 140 (vs. <140) (IU/mL) | 2.457 | 1.259 | 4.795 | 0.008 | | | | |
| ALP_rate $\geq$ $-47$ (vs. <$-47$) (%) | 0.393 | 0.193 | 0.797 | 0.01 | | | | |
| Ca $\geq$ 9.6 (vs. <9.6) | 1.456 | 0.754 | 2.813 | 0.264 | | | | |
| Ca_rate $\geq$ $-1$ (vs. <$-1$) (%) | 1.095 | 0.552 | 2.172 | 0.794 | | | | |
| Hb $\geq$ 11.4 (vs. <11.4) (g/dL) | 0.248 | 0.123 | 0.499 | <0.001 | 0.595 | 0.137 | 2.585 | 0.488 |
| Hb_rate $\geq$ $-8$ (vs. <$-8$) (%) | 0.401 | 0.205 | 0.787 | 0.008 | 0.236 | 0.087 | 0.643 | 0.005 |
| LDH $\geq$ 240 (vs. <240) (U/L) | 2.181 | 1.106 | 4.303 | 0.024 | | | | |
| LDH_rate $\geq$ 10 (vs. <10) (%) | 2.26 | 1.16 | 4.41 | 0.016 | | | | |
| Prior use of taxane (vs. no) | 2.605 | 1.297 | 5.234 | 0.007 | | | | |
| Prior use of abiraterone (vs. no) | 2.530 | 1.163 | 5.502 | 0.019 | | | | |
| Prior (or concurrent) use of abiraterone (vs. no) | 2.305 | 1.074 | 4.945 | 0.032 | | | | |
| Prior use of enzalutamide (vs. no) | 1.425 | 0.685 | 2.967 | 0.343 | | | | |
| Prior use of denosumab (vs. no) | 0.516 | 0.265 | 1.004 | 0.051 | | | | |

HR: hazard ratio; CI: confidence interval; BSI: bone scan index.

Log-rank test and Cox univariable analysis showed that several factors were significantly associated with prognosis (Figure 2). Among them, the median OS (95% confidence interval (CI)) was 32.6 months (19.0-NA) and 9.7 months (5.9–24.0) in the Early treatment group ($\leq$3rd line) and in the Late treatment group ($\geq$4th line) (*p* = 0.007), and 39.6 months (24.0-NA) in the BSI_low group (baseline BSI of <0.61) and 9.9 months (6.1–20.4) in the BSI_high group (baseline BSI of $\geq$0.61) (*p* < 0.001). The hazard ratio (HR) (95% CI) for the Late treatment group was 2.43 (1.25–4.75, *p* = 0.010) compared to the Early treatment group. In multivariable analysis, number of doses (6 vs. $\leq$5), PSA_rate ($\geq$45% vs. <45%), and Hb_rate ($\geq$$-8$ vs. <$-8$%) were significantly associated with HR to death, at 0.115 (0.026–0.943, *p* = 0.043), 6.052 (1.671–21.91, *p* = 0.006), and 0.236 (0.087–0.643, *p* = 0.005), respectively. There was no significant difference in OS from the first treatment line of mCRPC to death between the Early and Late groups (*p* = 0.549).

Next, we divided the above factors into two groups, BSI_high (baseline BSI of $\geq$0.61) vs. BSI_low (baseline BSI of <0.61) and Early ($\leq$3rd line treatment) vs. Late ($\geq$4th line treatment), and examined their causal relationships with the other factors using the Mann–Whitney U test (Tables 3 and 4). BSI data before Ra-223 administration (baseline BSI) were obtained from 53 of 64 patients. PSA and ALP levels at baseline were significantly higher in the BSI_high group than in the BSI_low group (*p* = 0.009 and *p* <0.001, respectively). In addition, the BSI_high group was also significantly associated with a lower Hb_rate (*p* = 0.015) and lower ALP_rate (*p* = 0.029). In the Early vs. Late comparison, the baseline PSA level was higher (*p* = 0.020) and the baseline Hb level was lower (*p* = 0.020) in the Late group. The Late group also received more prior treatments. The frequency of pretreatment in the Early and Late groups is shown in Table 5. In the Late group, 24/27 (88.9%), 20/27 (74.1%), and 18/27 (66.7%) patients had prior taxane, enzalutamide, and abiraterone use, respectively. The odds ratio for taxane use frequency was 4.1 times higher in the Late group

than in the Early group, and similarly 1.1 and 1.0 for abiraterone and enzalutamide groups, respectively.

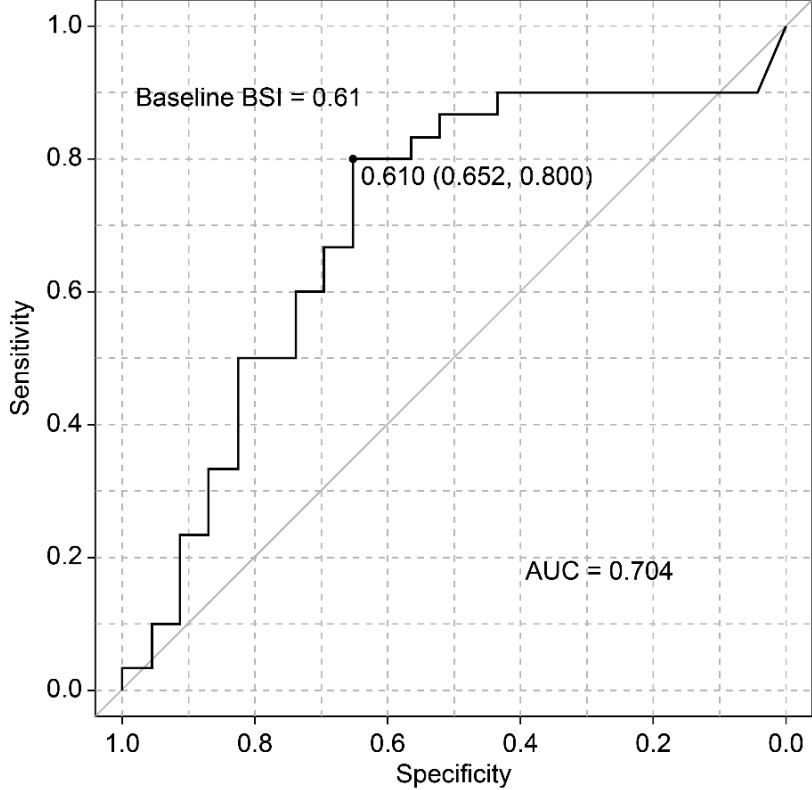

**Figure 1.** The ROC curve of baseline bone scan index. The Youden index and the left upper angle nearest point of the ROC curve were both 0.61, which was determined as the cutoff point of BSI. AUC: area under the ROC; ROC: receiving operator characteristic curve.

**Table 3.** Comparison between 2 groups (BSI_low vs. BSI_high) (MWU test).

| | BSI_Low (<0.61), n = 21 | | | BSI_High ($\geq$0.61), n = 32 | | | |
| --- | --- | --- | --- | --- | --- | --- | --- |
| | Median | IQR (0.25) | IQR (0.75) | Median | IQR (0.25) | IQR (0.75) | *p* |
| PSA | 13.06 | 5.52 | 19.40 | 58.45 | 9.15 | 152.80 | 0.009 |
| PSA_rate | 76.3% | −5.8% | 281.0% | 163.2% | 41.2% | 343.4% | 0.224 |
| PSADT | 50.48 | 35.31 | 142.39 | 59.82 | 43.75 | 169.53 | 0.430 |
| ALP | 82.00 | 59.00 | 113.03 | 163.65 | 103.00 | 291.55 | <0.001 |
| ALP_rate | −2.7% | −28.9% | 9.0% | −32.9% | −56.8% | −3.6% | 0.015 |
| LDH | 195.00 | 174.00 | 233.00 | 219.50 | 192.75 | 292.50 | 0.064 |
| LDH_rate | −3.1% | −10.9% | 12.3% | 12.1% | −1.9% | 25.7% | 0.093 |
| Ca | 9.40 | 9.10 | 9.50 | 9.40 | 8.98 | 9.70 | 0.750 |
| Ca_rate | 0.00 | −0.03 | 0.05 | 0.01 | −0.03 | 0.05 | 0.906 |
| Hb | 12.00 | 10.90 | 13.40 | 11.40 | 10.55 | 12.53 | 0.167 |
| Hb_rate | −2.2% | −10.4% | 4.7% | −7.9% | −27.5% | −1.5% | 0.030 |

MWU: Mann–Whitney U; IQR: interquartile range; BSI: bone scan index; PSA: prostate-specific antigen; PSADT: PSA doubling time; ALP: alkaline phosphatase; LDH: lactate dehydrogenase; Ca: calcium; Alb: albumin; Hb: hemoglobin; Neut: neutrophil; Plt: platelets.

The time to pain progression is shown in Figure 3. The median (95%CI) p-PFS of the BSI_low group and BSI_high group was 41.9 (27.9-NA), and 15.5 (6.9–26.0) months (*p* = 0.002), respectively, and for the ALP_low group (baseline ALP of <140 IU/mL) and ALP_high group (baseline ALP of $\geq$140 IU/mL) was 41.9 (21.7-NA) and 15.5 (9.6-NA) months (*p* = 0.010), respectively. In comparison, there was no significant difference between those with prior use and no prior use of denosumab; however, patients in the denosumab use group did not reach the median p-PFS (Figure 3d).

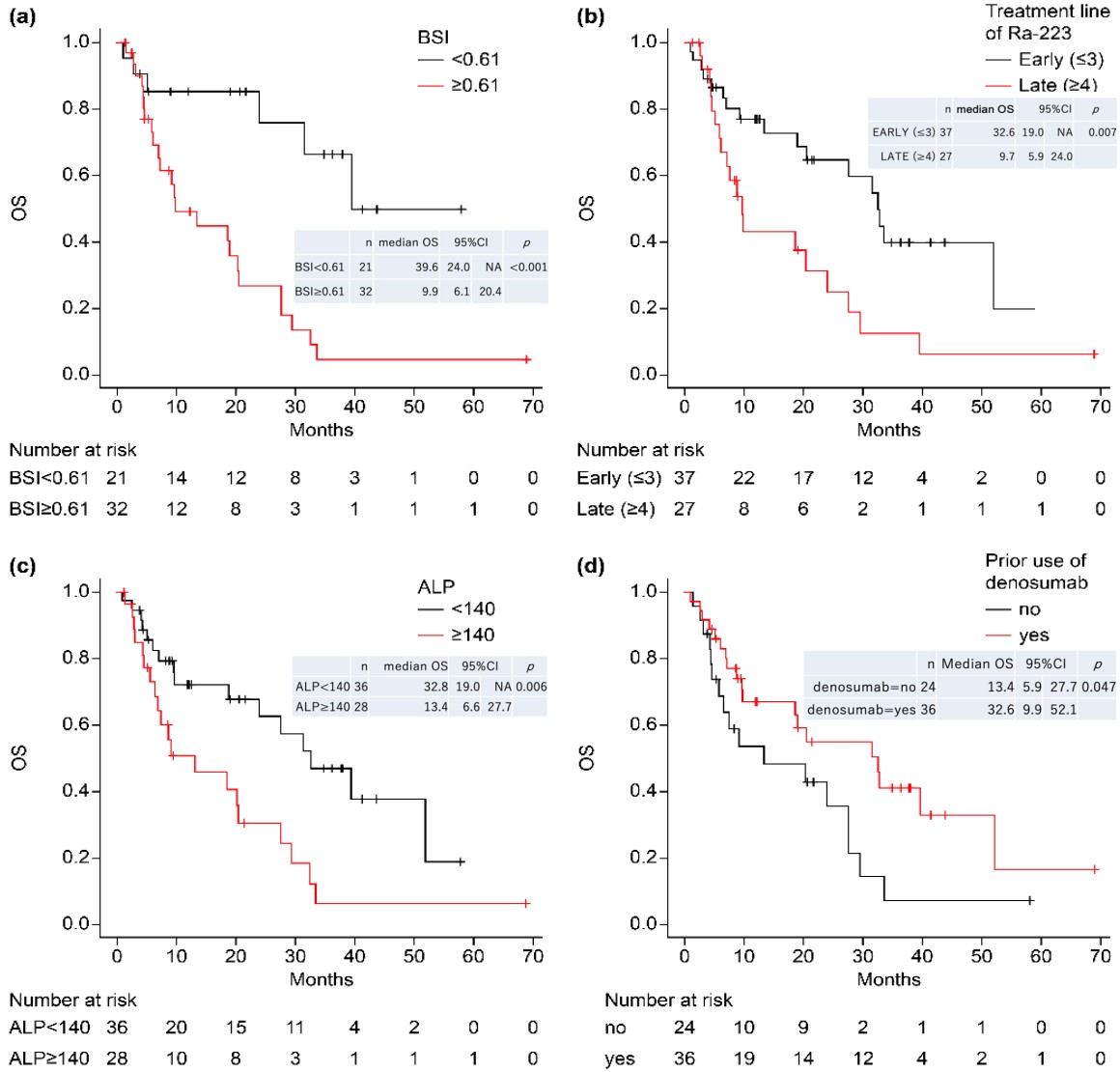

**Figure 2.** Log-rank test of OS comparison: (**a**) BSI<0.61 vs. BSI ≥ 0.61; (**b**) treatment line of Ra-223, Early (≤3) vs. Late (≥4); (**c**) ALP < 140 vs. ALP ≥ 140; (**d**) prior use of denosumab, no vs. yes. Median OS was significantly longer in patients with baseline BSI of <0.61, early treatment (≤3rd line), baseline ALP of <140U/l, and prior use of denosumab. OS: overall survival. NA: not available.

**Table 4.** Comparison between 2 groups of treatment line (Early vs. Late) (MWU test).

| | Early (≤3rd Line), n = 37 | | | Late (≥4th Line), n = 27 | | | |
|---|---|---|---|---|---|---|---|
| | Median | IQR (0.25) | IQR (0.75) | Median | IQR (0.25) | IQR (0.75) | $p$ |
| PSA | 13.06 | 5.73 | 97.56 | 59.10 | 19.00 | 92.10 | 0.021 |
| PSA_rate | 97.5% | 1.6% | 320.2% | 188.0% | 45.0% | 346.2% | 0.179 |
| ALP | 119.00 | 77.00 | 208.80 | 110.92 | 58.90 | 227.46 | 0.331 |
| ALP_rate | −27.8% | −57.1% | −2.7% | −9.6% | −39.6% | 7.9% | 0.087 |
| LDH | 214.00 | 181.00 | 251.00 | 213.00 | 187.50 | 292.00 | 0.446 |
| LDH_rate | 2.2% | −10.0% | 24.9% | 12.5% | −4.1% | 32.9% | 0.224 |
| Hb | 12.50 | 11.30 | 13.00 | 11.00 | 10.50 | 12.20 | 0.020 |
| Hb_rate | −7.9% | −13.3% | 0.0% | −6.3% | −19.3% | 2.4% | 0.961 |
| BSI | 0.98 | 0.22 | 3.73 | 0.93 | 0.44 | 1.93 | 0.979 |

MWU: Mann–Whitney U; IQR: interquartile range; BSI: bone scan index; PSA: prostate-specific antigen; PSADT: PSA doubling time; ALP: alkaline phosphatase; LDH: lactate dehydrogenase; Ca: calcium; Alb: albumin; Hb: hemoglobin; Neut: neutrophil; Plt: platelets

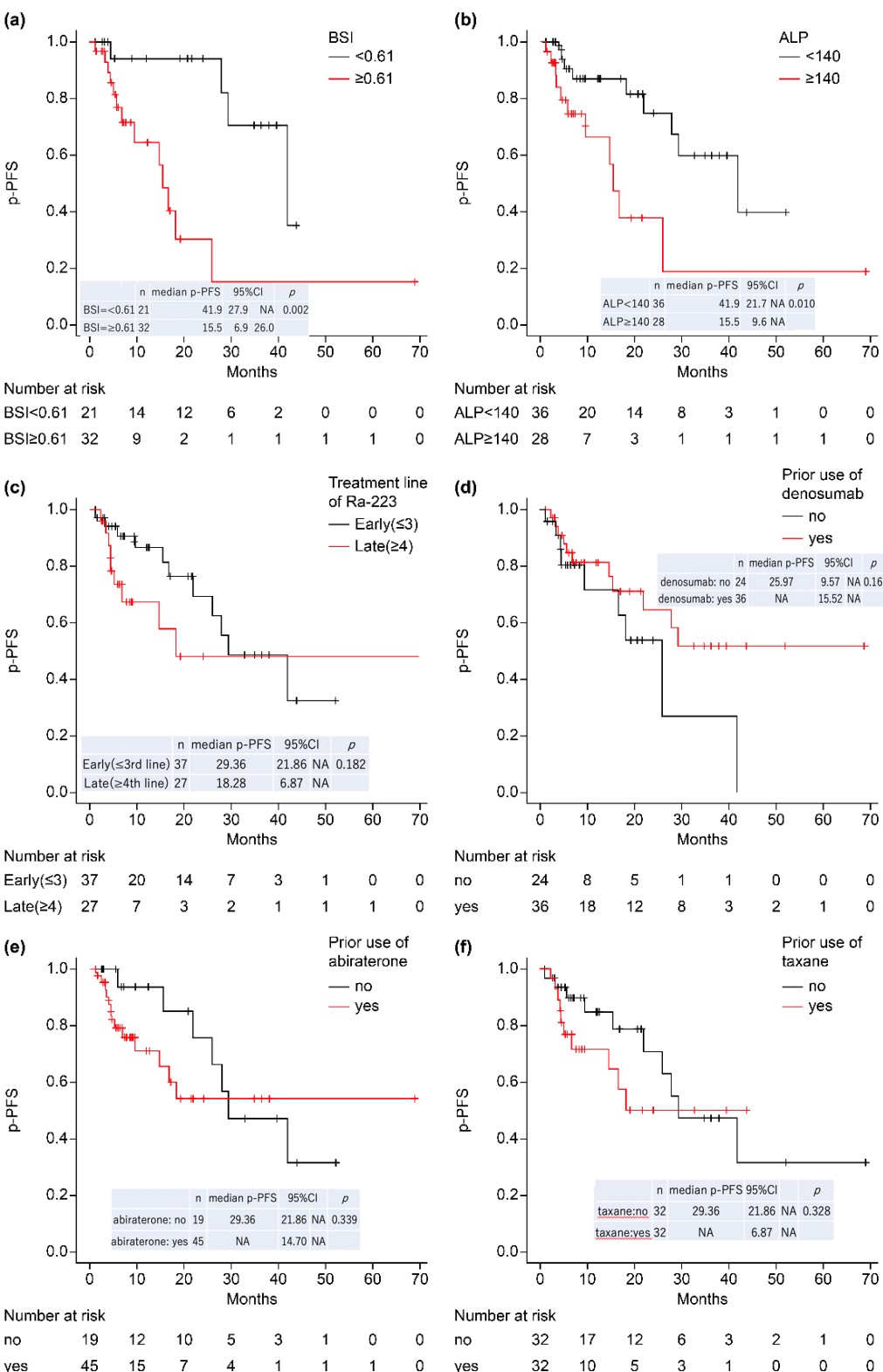

**Figure 3.** Log-rank test of p-PFS: (**a**) BSI < 0.61 vs. BSI $\geq$ 0.61; (**b**) ALP < 140 vs. ALP $\geq$ 140; (**c**) treatment line of Ra-223, Early ($\leq$3) vs. Late (4$\geq$); (**d**) prior use of denosumab, no vs. yes; (**e**) prior use of abiraterone, no vs. yes; (**f**) prior use of taxane, no vs. yes. Figure 3 (**c**–**f**) shows that p-PFS is prolonged in the early treatment group ($\leq$3rd line) and in the prior use of denosumab group, while p-PFS is shortened in the prior use of abiraterone and taxane groups, although the differences are not significant. p-PFS: pain progression-free survival. NA: not available.

There was no significant difference in the p-PFS between patients with and without pain at the time of the first dose of Ra-223. Between ALP and ALP_rate, Spearman's correlation coefficient was −0.521. The median (range) NLR obtained from 48 cases were 3.05 (1.19–26.11) respectively. A preliminary study of the relationship between NLR and prognosis did not show significant differences.

**Table 5.** Frequency of pretreatment type in Early and Late group.

| Pretreatment Type | Early (n = 37) | Late (n = 27) | OR (Late/Early) |
|---|---|---|---|
| Taxane (n = 32) | 8/37 (25%) | 24/27 (88.9%) | 4.1 |
| Abiraterone (n = 45) | 25/37 (67.6%) | 20/27 (74.1%) | 1.1 |
| Enzarutamide (n = 43) | 25/37 (67.6%) | 18/27 (66.7%) | 1.0 |

OR: odds ratio.

## 4. Discussion

To our knowledge, this is one of the largest studies conducted at a single institution in Japan regarding the outcomes and prognostic factors of Ra-223 therapy that even included asymptomatic patients. According to the National Comprehensive Cancer Network Guidelines 2022 [25], docetaxel, abiraterone, and enzalutamide are listed as first-line agents for the treatment of mCRPC and Ra-223 is indicated only for mCRPC with pain symptoms, and this is also stated by the U.S. Food and Drug administration [26] and European Medicines Agency [27]. In Japan, however, Ra-223 is also indicated for asymptomatic mCRPC patients [28,29], who were also included in this study. This could clarify the role of Ra-223 regarding pain management in a clinical situation different from that in the USA and Europe. Accordingly, we could assess p-PSF in addition to OS. This p-PSF, which is one of the SSEs related to the efficacy assessment of Ra-223, may represent a good index of the QOL of patients. We think it is meaningful that the prognostic factors were similar to those reported in previous studies, even when asymptomatic patients were included [16,18]. In short, the initial PSA and N stage at diagnosis did not affect prognosis; instead, T4 and a Gleason score of 10 at diagnosis were factors that worsened the prognosis. The use of denosumab improved prognosis and prolonged p-PFS. LDH was also shown to be an important biomarker, which, to our knowledge, is a new finding from Japanese studies. Although the subgroup analysis of ALSYMPCA showed that the change in PSA from baseline to 12 weeks after randomization of Ra-223 was not related to prognosis [12], this study could show that changes in PSA (PSA_rate) were significantly associated with prognosis. This difference may have occurred due to the longer follow-up period of 24 weeks (during six cycles of Ra-223 therapy), which is a new finding as far as we know.

### 4.1. Relationship between BSI and Other Factors

We observed a similar relationship between BSI and the prognosis, as shown in a previous study [30]. Some previous reports have shown that the intensity of bone metastasis in terms of the number of bone metastases or super-scan on bone scintigraphy is associated with prognosis [12,13,16,18]. In this report, we showed that BSI_low (<0.61) was associated with significantly higher OS and that BSI_high was associated with high baseline PSA and ALP levels. In addition, BSI_high was associated with a low Hb_rate and low ALP_rate. These results are consistent with the subgroup analysis of the ALSYMPCA trial, in which a higher number of bone metastases was associated with a higher risk of anemia, and Ra-223 was effective even with more bone metastases [9]. It is important to note that anemia was not associated with Ra-223 use [8]. Although ALP and BSI at baseline appeared to be related biomarkers, the area under the ROC curve comparisons were 0.601 and 0.704, respectively, suggesting that BSI may be a better prognostic biomarker.

*4.2. Relationship between Baseline ALP and ALP_Rate*

This study showed that ALP_rate ($<-47\%$) alone significantly worsened OS. However, ALP and ALP_rate were negatively correlated with a Spearman's rank correlation coefficient of $-0.521$, suggesting that ALP_rate was a confounding factor.

*4.3. Relationship between Treatment Line and Prognosis*

The relationship between the treatment line (Early or Late) and the prognosis is discussed. The earlier the treatment, the better the prognosis, which is consistent with a previous report from Japan [18]. The results of the comparison of the Early group ($\leq$3rd line) and Late group ($\geq$4th line) showed that baseline PSA was higher and baseline Hb was lower in the Late group ($p = 0.020$, $p = 0.020$, respectively); this indicated that the Late group had more pretreatment, but baseline PSA seemed to increase over time, even with pretreatment. Of 27 patients in the Late group, 24 (88.9%) had a history of taxane use. The OS was also significantly worse in those with prior use of abiraterone. These results may have contributed to the poor prognosis in the Late group and suggest that Ra-223 may be used before abiraterone treatment. However, there was no significant difference in prognosis with prior use of enzalutamide. For anticancer agents, it seems that Ra-223 should be used before taxane treatment. Moreover, there was no significant difference in OS from the first line treatment after the diagnosis of mCRPC between the Early and Late treatment groups. This is thought to be because the treatment was tailored to each patient over a long period of time, and therefore, no significant difference was observed.

*4.4. Time to Pain Progression*

In this report, the time to pain progression was defined as the time to the initiation or addition of analgesics or opioids from the first Ra-223 administration. Median p-PFS was significantly prolonged in the BSI_low and ALP_low groups. Although there were no significant differences, early administration of Ra-223 and prior use of denosumab prolonged p-PFS, and prior use of taxane and abiraterone resulted in shorter p-PFS. These results indicate that the early administration of Ra-223 is associated with improved QOL, similar to the subgroup analysis of ALSYMPCA [31]. In terms of pathological fracture, no pathological fractures were observed in this study, even in the patients who received abiraterone, which was different from the results reported in the ERA223 study and remain to be solved [20].

*4.5. Limitations*

This retrospective study analyzed the results of a relatively small number of patients. However, this study was conducted at a single institution; thus, the treatment background is somewhat uniform, which might reduce the heterogeneity of patient management. In addition, the factors identified in this study were prognostic factors from the date of Ra-223 initiation, but they were not identified as prognostic factors after the start of the entire treatment. Although the present study was able to demonstrate some causal relationships among the factors, various other factors may have a complex relationship. Further investigations are needed to validate our results in a larger-sized study.

**5. Conclusions**

In this study, several factors were associated with OS and p-PFS, including baseline BSI, ALP, LDH, and Hb, and prior use of denosumab, even when asymptomatic patients were included. In addition, early administration of Ra-223 was significantly associated with improved OS, and it may be meaningful to administer Ra-223 before novel hormonal or anticancer agents. Summarily, it is important to carefully consider the patient's condition and the acceptability of treatment based on BSI, ALP, LDH, and Hb, and to determine the optimal sequence and combination of other therapies, such as a novel hormonal agent and anticancer agents, in addition to Ra-223, to improve not only OS but also p-PFS.

**Author Contributions:** Conceptualization, Y.O., M.H. and K.O.; methodology, M.H. and K.O.; software, Y.O.; validation, M.H. and K.O.; formal analysis, Y.O. and M.H.; investigation, Y.O., M.H., M.I., T.U. and K.O.; resources, K.Y., H.U., Y.W. and Y.Y.; data curation, Y.O., A.R. and E.I.; writing—original draft preparation, Y.O.; writing—review and editing, all authors.; visualization, Y.O.; supervision, M.H., K.O., K.N., H.D., T.M. and Y.N.; project administration, Y.O. and M.H.; All authors have read and agreed to the published version of the manuscript.

**Funding:** This work was supported by the Japan Society for the Promotion of Science Grants-in-Aid for Scientific Research (JSPS KAKENHI), Grant Number JP21K07576.

**Institutional Review Board Statement:** The study was conducted according to the tenets of the Declaration of Helsinki. Institutional review board approval was obtained from the Ethics Committee of the Kindai University Faculty of Medicine (R02-308) and Nara Hospital, Kindai University (671).

**Informed Consent Statement:** Written informed consent for Ra-223 therapy was obtained from all individual participants prior to treatment. Informed consent for this study was obtained in an opt-out form on the website.

**Data Availability Statement:** The data are available from the corresponding authors upon reasonable request.

**Conflicts of Interest:** The authors declare no conflict of interest.

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
