# Peer review of "Investigation into the Optimal Strategy of Radium-223 Therapy for Metastatic Castration-Resistant Prostate Cancer"

_radiation, doi:10.3390/radiation2030021_

Round 1

Reviewer 1 Report (Previous Reviewer 3)

The textual improvements of the manuscript allowed a better understanding of the study objective and its execution compared with the previous version. However, this did not alter the study's outcome and still does not show what is novel regarding prognostic biomarkers compared with other studies with Ra-233 in mCRPC.

Author Response

 In Japan, unlike in Europe and the U.S., Ra-223 treatment is indicated even for asymptomatic patients (as described in Discussion), and we think it was meaningful to confirm the same prognostic biomarkers that had been reported so far in a group including asymptomatic patients. Furthermore, since Ra-223 is reported to be effective in maintaining QOL, it is important to examine the progression of pain at the same time. We revised the discussion and conclusion to emphasize this.

 Although the sub-group analysis of ALSYMPCA showed that the change in PSA from baseline to 12 weeks after randomization of Ra-223 was not related to prognosis (Sartor, Ann. oncol 2017,[12]), this study could show that changes in PSA (PSA_rate) were significantly associated with prognosis. This difference may have occurred due to the longer follow-up period of 24 weeks (period during 6 cycles of Ra-223 therapy) which is a new finding as far as we know. We added this to the discussion.

 We think it is also new that this study investigated the previously reported prognostic factors more deeply by searching for causal relationships. In particular, we focused on BSI and treatment line, and found that high BSI was associated with ALP and PSA as expected, but we also found that Hb_rate was significantly lower in high BSI group, which we think is an important point related to the acceptability of treatment.

 Although there have already been reports on BSI, treatment line, and prognosis, as described above, to the best of our knowledge, there have been few reports that showed why each of these factors affects prognosis. (lines 36-40)

Reviewer 2 Report (New Reviewer)

Authors have to clarify the indication of radium 223 treatment in Japanese patients to compare with EMA and FDA indication

The results about OS in patients with prior use of denosumab should be improved, because there are no data about the influence of denosumab on OS

Authors could consider tha addition inflammatory biomarkers (nlr) among prognostic factors

Author Response

 We have confirmed that Ra-223 treatment is only for symptomatic mCRPC patients according to FDA and EMA guidelines. We have added that to the discussion and the reference sections.

https://www.ema.europa.eu/en/medicines/human/referrals/xofigo

 The difference in OS with and without denosumab was as described in fig 2.(d). The median OS (months, 95%CI) of not use group vs denosumab use group was 13.4 (5.9-27.7M) vs 32.6(9.9-52.1M), p=0.047 respectively. In the Cox univariate analysis, the result was described in the last row of the Table 2. And the difference was not significant (p=0.051), which we considered too complicated to describe in the text of result, so we limited our description to figures and tables.

In this study, we confirmed that the neutrophil count was not significantly different as a prognostic factor. We conducted a preliminary study of NLR from 48 cases based on your suggestion and found no significant differences too. (Median, min-max were 3.05, 1.19-26.11 respectively.) So, we did not include them in this paper. Thank you very much again for your valuable suggestion.

Reviewer 3 Report (New Reviewer)

Overall survival (OS) and pain progression-free survival (p-PFS),M which was proposed as a measure of quality of life (QOL), so in line 281 instead of QOL the PFS is more appropriate.

Lines 81-84:  how many patients have MR done? please explain more clearly the concept of bone scan index (BSI) and why 0.61 was taken as a reference value. 

Author Response

According to the comment, line 281 was revised as “p-PFS” from “QOL”.

Bone metastases were diagnosed on the basis of bone scintigraphy in all of the patients. Nine patients had an adjunctive MRI at the time of diagnosis of bone metastasis.

We added the explanation of BSI to the Materials and methods.

The cut-off point for BSI was obtained from the Youden index (it was also equal to the left upper angle nearest point in this case) of the ROC curve for mortality events. (Lines 131-132, figure 1)

Round 2

Reviewer 1 Report (Previous Reviewer 3)

The changes in the latest version of the manuscript better explain the set-up of the study, similarities and differences in treatment of mCRPC patients in Japan, Europe, and USA, and the identified prognostic factors.

This manuscript is a resubmission of an earlier submission. The following is a list of the peer review reports and author responses from that submission.

Round 1

Reviewer 1 Report

The manuscript describes a retrospective cohort of 64 mCRPC patients treated with Ra-223 at a single institution in Japan with a median follow-up time of 9.6 months. They reported overall survival and pain progression-free survival outcomes in their cohort and examined the prognostic value of various clinical factors and biomarkers. They identified several variables associated with better outcomes, including early treatment (≤3rd line), completion of 6 cycles, lower bone scan index, lower ALP, lower PSA, lower LDH, higher Hb, and prior denosumab use.

Overall, this study is not very novel and does not provide much new information beyond what is already reported in the literature regarding Ra-223 biomarkers. All of the prognostic factors and biomarkers described in this manuscript have already been previously reported in prior studies as associated with outcomes after Ra-223 therapy. As such, the data provided is mostly confirmatory and does not really advance the current knowledge of the field. In addition, there are several problems with the study design and manuscript as follows:

-       This is a retrospective study of a rather heterogeneous cohort, and all of the prognostic factors identified are potentially suspect due to selection bias. This would have been potentially acceptable if the study had focused on novel biomarkers. However, because the study focuses on previously reported known biomarkers (many of which have already been studied within prospective trials), my enthusiasm for this study is further decreased.

-       The median follow-up time for this study is very short at 9.6 months, compared to the median OS of 14.9 months reported in the ALSYMPCA trial. Thus, the follow-up time of the study is too short to provide reliable data on overall survival endpoints.

-       All K-M survival curves in Figures 2 and 3 display X-axes through 70 months, even though the median follow-up time is only 9.6 months. K-M curves should be cut off on the X-axis at a time close to the median follow-up time to avoid displaying unreliable data from small numbers of patients at risk at later time points.

-       Toxicity and patient-reported quality of life outcomes data are not provided. This information would be critical to provide given the authors’ stated intention in the Abstract “to improve OS and quality of life…”

-       Performance status was not included as a patient characteristic.

-       Patient medical comorbidities were not included within patient characteristics.

-       “Multivariate” analysis should be described as “multivariable” analysis.

Reviewer 2 Report

The authors provide results of a retrospective analysis of factors that may predict OS / pain-PFS in a small group of mCPRC patients who received 1-6 cycles of radium-223.

Major comments

- Limited number of patients, retrospective nature

- An extensive number of prognostic factors is evaluated (29!)

- Data on the univariate and multivariate analysis of pPFS is missing.

- Since OS is discussed: the authors fail to provide information on the treatments that were still given AFTER radium-treatment.

- The division in early and late is arbitrary. Taxane regimens may or may not be part of both groups. It is unclear whether these lines of treatment only include mCPRC lines.

- The authors seem to choose prognostic factors to investigate further, rather than only using those which are significant in multivariate analysis. F.e. treatment line of radium-223 is investigated further, however, this is not a significant prognosticator in the multivariate analysis. On the other hand, no further information/discussion is provided regarding PSA, PSA-rate, Hb and Hb-rate.

- Wording in the abstract is not appropriate: early radium treatment is associated with longer OS (which is not validated in the multivariate analysis however, so should not be detailed in the conclusion of the abstract), but this does not mean that early radium treatment prolongs OS. The detected correlations may be caused by other factors, such as the general state of the patient (more extensive disease, worse clinical condition,...).

- The abstract does not correctly represent the data of the paper. It is stated that 'univariate and multivariate analyses revealed early treatment, 6 treatment-cycles completion, low BSI, low ALP, low PSA, low LDH, high Hb, prior denosumab'. This is not correct.

Minor comments:

- Alpha radiation is formed by alpha particles, these are not rays.

- Were all patients taking bone supportive agents during radium treatment?

- There is an inconsistency between the number of patients with LN metastases between the text and table 1.

- The title of table 2 is not clear. Are these the results of the OS analysis?

- Figure 2: it is not clear why these parameters were chosen and shown, since 3/4 not significant in multivariate analysis.

- Was BSI 0.61 also optimal cut-off for p-PFS?

Reviewer 3 Report

In this retrospective study, based on a single institution's data, the 

authors aim to identify prognostic factors after Ra-223 administration 

in Japanese metastatic castration-resistant prostate cancer patients and

 to determine an optimal treatment strategy. For that, they explored 

potential statistical correlations between overall survival (OS) and 

pain progression-free survival (p-PFS) on the one hand, and blood 

parameters, bone scan index, and the number of treatment lines before 

RA-223 administration, on the other hand. The results showed that early 

administration of Ra-223 (already described in the literature) and bone 

scan index are the only strong prognostic factors for OS and p-PFS. 

Missing is a discussion of:

  1. the differences in treatment (guidelines) of metastatic castration-resistant prostate cancer patients in Japan and, e.g., Europe and the USA, and the meaning of this in the interpretation of their results internationally,
  2. Other reported clinical studies in the same patient population also explored prognostic biomarkers but, apart from one reference, other reports are ignored,
  3. (inverse) relationship between disease progression and blood parameters such as ALP, LDH, and Hb.